# Recent Advances in Imaging Macular Atrophy for Late-Stage Age-Related Macular Degeneration

**DOI:** 10.3390/diagnostics13243635

**Published:** 2023-12-10

**Authors:** Anny M. S. Cheng, Kakarla V. Chalam, Vikram S. Brar, David T. Y. Yang, Jineel Bhatt, Raphael G. Banoub, Shailesh K. Gupta

**Affiliations:** 1Department of Ophthalmology, Broward Health, Fort Lauderdale, FL 33064, USA; annycheng0927@gmail.com (A.M.S.C.); rbanoub@browardhealth.org (R.G.B.); 2Specialty Retina Center, Coral Springs, FL 33067, USA; docjineel@gmail.com; 3Department of Ophthalmology, Herbert Wertheim College of Medicine, Florida International University, Miami, FL 33199, USA; 4Department of Ophthalmology, Loma Linda University, Loma Linda, CA 92350, USA; kakarla.chalam@gmail.com; 5Department of Ophthalmology, Virginia Commonwealth University, Richmond, VA 23298, USA; vikram.brar@vcuhealth.org; 6College of Biological Science, University of California, Davis, Sacramento, CA 95616, USA; davidyang0709@gmail.com

**Keywords:** age-related macular degeneration, confocal scanning laser ophthalmoscope, fundus autofluorescence, macular atrophy, microperimetry, multifocal electroretinogram, optical coherence tomography angiography, optical coherence tomography

## Abstract

Age-related macular degeneration (AMD) is a leading cause of blindness worldwide. In late-stage AMD, geographic atrophy (GA) of dry AMD or choroidal neovascularization (CNV) of neovascular AMD eventually results in macular atrophy (MA), leading to significant visual loss. Despite the development of innovative therapies, there are currently no established effective treatments for MA. As a result, early detection of MA is critical in identifying later central macular involvement throughout time. Accurate and early diagnosis is achieved through a combination of clinical examination and imaging techniques. Our review of the literature depicts advances in retinal imaging to identify biomarkers of progression and risk factors for late AMD. Imaging methods like fundus photography; dye-based angiography; fundus autofluorescence (FAF); near-infrared reflectance (NIR); optical coherence tomography (OCT); and optical coherence tomography angiography (OCTA) can be used to detect and monitor the progression of retinal atrophy. These evolving diverse imaging modalities optimize detection of pathologic anatomy and measurement of visual function; they may also contribute to the understanding of underlying mechanistic pathways, particularly the underlying MA changes in late AMD.

## 1. Introduction

Age-related macular degeneration (AMD) has been recognized as one of the leading causes of vision impairment and blindness in the elderly worldwide [1]. In a meta-analysis of individuals aged 45–85 years old, the pooled global prevalence of late, and any stage of AMD was 0.37%, and 8.69%, respectively. The number of individuals globally with AMD is projected to increase from 196 million in 2020 to 288 million in 2040 [2]. In the late stage of AMD, significant visual impairment is due to dry AMD with geographic atrophy (GA), or wet AMD with choroidal neovascularization (CNV). In GA, associated with dry AMD, the retinal pigment epithelium, choriocapillaris, and photoreceptors are progressively atrophied [3]; in wet neovasular AMD (nAMD), choroidal neovascularization penetrates Bruch’s membrane leading to the leakage of fluid, lipid, and blood, resulting in retinal fibrosis. Macular atrophy (MA), an anatomic endpoint of AMD representing both GA in dry AMD and CNV in nAMD, is characterized by the permanent degradation of the retinal pigment epithelium (RPE) and overlying photoreceptors. Multiple studies have shown that natural progression to MA is the final common pathway in both nAMD and dry AMD disease progression [4]. Due to its chronic nature and progressive increase in vision loss, AMD, and specifically MA, will continue to be a global public health concern with substantial socioeconomic and healthcare consequences.

The conventional classification of AMD relies on clinical examinations such as the visual acuity test, the Amsler grid test, dilated ophthalmoscopy, or color fundus photography (CFP) [5]. With developments in retinal imaging, a variety of diagnostic imaging techniques are now accessible to help establish different stages of AMD. The utilization of optical coherence tomography (OCT), angiography, and other novel multimodal imaging techniques has provided access to the histological details of AMD, revealing previously unknown anatomical characteristics (reviewed in [6]). Many of these characteristics, which pose a threat to vision, are correlated with the likelihood of developing late AMD. 

In late-stage AMD, MA is associated with reticular pseudodrusen (RPD), (located subretinally as yellowish-white net-like patterns), which is a risk factor for progression to the late stage of AMD in both dry and wet AMD patients [7,8]. MA forms in areas previously occupied by drusenoid pigment epithelial detachments (PED) and is characterized by confluent loss of the RPE [9]. It has been proposed that MA development may depend on the underlying MA phenotype, in which type 3 retinal angiomatous proliferation (RAP) lesions may have a greater risk of development and progression of atrophy, whereas type 1 lesions are associated with a lower risk of MA progression [10,11]. Currently, there are no standard and effective treatments for MA despite emerging innovative therapies. With new therapeutics on the horizon, choosing patients for trials or newly developed therapies that will improve the clinical course of AMD as well as early detection, may help delay or halt disease progression. Hence, it is important to detect incidental MA at the first appearance to identify subsequent progression to central macular involvement over time. Different detection and imaging modalities are evolving for identifying disease progression and prognostic factors; these may also contribute to the understanding of pathogenetic pathways, specifically the underlying macular changes in late AMD. A summary of these methods is provided below.

## 2. Clinical Assessment

### Clinical Presentation and Examinations

Early asymptomatic AMD is typically diagnosed based on the patient’s age and a comprehensive dilated eye examination for characteristic signs such as drusen or retinal pigment changes. The progression of AMD to advanced stages invariably involves the foveal region, which develops dense and irreversible scotomas, resulting in retinal function impairment and irreversible vision loss. The Amsler grid can help patients monitor their changes in central vision distortions. Historically, the progression of visual impairment and the estimation of ultimate residual visual function are determined by measuring visual acuity. Standard visual acuity tests, such as best corrected visual acuity (BCVA), do not fully capture the functional impact of atrophic AMD because lesions frequently spare the foveal center in the early stage, causing standard vision charts to falsely indicate that vision is unaffected [12]. Other tests, such as dark adaptation, flicker threshold, and photostress recovery time are more sensitive than BCVA in detecting early functional loss in AMD [13,14,15]; however, they are time-consuming and therefore limited in their clinical use. The ophthalmic electrophysiology test, electroretinogram (ERG), can detect the functional abnormalities observed in AMD, such as the early loss of rod photoreceptors and the loss of central and paracentral perimetric sensitivities. Multiple studies have reported the efficacy and accuracy of multi-focal ERG (mfERG) in detecting photoreceptor degeneration and macular function disturbances, which may be beneficial in the early diagnosis and progression of AMD [16,17,18,19,20,21,22,23,24,25,26,27].

## 3. Assessment Using Imaging

Manifesting as vision loss as a lesion enlarges and encompasses the foveal center, MA represents the final morphological stage of a degenerative disease process [28]. The course of MA in late AMD is characterized by the development of atrophic areas that enlarge continuously over time with cell death of the RPE, the outer neurosensory retina, and the choriocapillaris [29]. As imaging technologies evolve, their diagnostic and monitoring applications for AMD and MA have expanded. Currently, anatomic assessment of AMD via multimodal fundus imaging is commonly used to diagnose and monitor the disease (Table 1).

### 3.1. Fundus Photography

Historically, CFP has been the benchmark for documenting funduscopic abnormalities. The conventional classification of AMD systems, the Age-Related Eye Disease Study (AREDS), relies on clinical examination, or CFP [5]. With its wide range of illumination, CFP is the imaging modality that most closely resembles clinical ophthalmoscopy. AMD-related phenotypic alterations (Figure 1A), including drusen, crystalline deposits, pigmentary changes, lipid, atrophy, and fibrosis, as well as neovascular findings including hemorrhages, fluid, and exudate, can be detected using CFP. However, CFP contrast is comparatively low, posing difficulties in the detection and measurement of atrophic lesions in comparison to alternative imaging modalities. Of note, RPD, as a major risk factor for progression to late AMD, is overlooked with standard CFP but was detected in more than a quarter of patients [30]. The precise delineation of atrophic lesion boundaries, especially in eyes with multifocal lesions that are relatively small, poses challenges in applying CFP to measure areas of MA and monitor their expansion over time [31]. The sensitivity required to identify topographic changes at the margins of atrophy zones can be enhanced using stereoscopic viewing of CFP. Nevertheless, the clinical applicability of stereoscopic image capture is limited by the need for experienced photographers and patient cooperation.

Recently, multicolor imaging has been developed [32]. Multicolor images consist of a composite image made using confocal scanning laser ophthalmoscopy (cSLO) to capture three simultaneous laser wavelengths: blue reflectance (486 nm), green reflectance (518 nm) and infrared reflectance (815 nm) [33]. These various wavelengths of light penetrate and reveal the details of different retinal layers. Blue reflectance, with the shortest wavelength, reaches the vitreoretinal interface and inner retina, whereas infrared reflectance penetrates the deepest to detect structures in the outer retina and choroid. One study reported that multicolor imaging detected small-sized atrophic AMD lesions [34]. However, retinal structures become less distinct due to chromatic aberration caused by the differential focal planes of the three different wavelength lasers. In addition, their optical reflection properties can hinder the distinction between subtle hemorrhages and pigmentary lesions. To date, only a few studies using this imaging are available [34,35,36]. Due to the limitations of current knowledge, the application of multicolor imaging for atrophic and nAMD should be optional, as its utility has yet to be demonstrated. 

Widefield imaging with a field of view that exceeds 100 degrees enables visualization of larger areas of the retina. A consensus group of retinal imaging experts defined ultra-widefield (UWF) as a single-capture image, centered on the fovea, which captures retinal anatomic features anterior to the vortex vein ampullae from 110° to 220° or 97% in all four quadrants. The utilization of UWF imaging has enabled the monitoring of macular and peripheral findings in both dry AMD (Figure 1B) and nAMD [37,38,39]. A meta-analysis showed that peripheral lesions including early or late retinal atrophic changes were identified in 82.7% of AMD eyes [39]. This finding suggested that AMD is, in fact, a pan-retinal and not just a macular disease. Although UWF can monitor peripheral abnormalities to provide a more complete understanding of AMD [38], the clinical significance of peripheral lesions in AMD remains incompletely understood. 

### 3.2. Dye-Based Angiography-Fluorescein and Indocyanine Green Angiography

Fundus fluorescein angiography (FA) utilizes sodium fluorescein dye to illuminate the retina at a peak wavelength of 490 nm (blue), and then photographically records the excited fluorescent 530 nm (green) light that is emitted [40]. In FA, atrophic patches appear as well-defined, hyperfluorescent areas due to enhanced visualization of the normal choroidal fluorescence caused by the loss of RPE cells (window defect), which would normally diminish the transmission of fluorescein fluorescence (Figure 1C). Compared to CFP, this demarcated hyperfluorescent signal provides a sharper contrast between the atrophic and the surrounding non-atrophic areas. However, other pathologic findings in dry AMD such as Drusen, or pigmentary changes, or fibrotic tissue in dry AMD; neovasularization in nAMD, may also result in an increased fluorescence signal or progressive dye leakage and therefore obscure the boundary demarcation of atrophy [41,42]. FA is therefore recommended for the detection, classification, and quantification of NV but not atrophic changes.

Indocyanine green angiography (ICG-A) utilizes ICG, a molecule that is 98% protein bound and therefore remains in the fenestrated choriocapillaris longer and leaks less, relative to fluorescein dye. It is sometimes more useful than fluorescein dye to study choroidal diseases such as nAMD [43]. In ICG-A, atrophic patches appear as discrete hypofluorescent areas with a loss of background fluorescence owing to small and medium vessel choriocapillaris atrophy [44]. However, the large, deep choroidal vessels may still be visible, interfering with the outline of the area of atrophy and causing more difficulty in exact and reliable delineation. While ICG-A is useful in distinguishing polypoidal choroidal vasculopathy (PCV); chronic central serous chorioretinopathy (CSC); and retinal angiomatous proliferation (RAP) from classic nAMD [45], ICG-A has a negligible role for the identification of atrophy in AMD. In addition, both ICG-A and FA are invasive procedures that carry the risk of local infiltration, extravasation at the injection site, and allergic reaction to the intravenously administered dye which, though rare, can be severe and life-threatening [46].

### 3.3. Fundus Autofluorescence

Fundus autofluorescence (FAF) is a non-invasive method that provides rapid, noninvasive, high-contrast retinal images that are particularly useful for detecting atrophic areas, and better atrophic lesion boundary discrimination when compared with CFP [47,48] (Figure 1D). FAF utilizes the fluorescent properties of lipofuscin, a byproduct of lysosomal breakdown of photoreceptor outer segments within the RPE cell. When excited by an appropriate light source, the bisretinoid components of lipofuscin absorb blue light with a peak excitation wavelength of approximately 470 nm, and emit yellow-green light with a peak wavelength of 600 nm. A detector is used to record the emissions signals as they are emitted. A FAF image, then, is a density map of lipofuscin where the brighter “hyperfluorescent” areas represent areas of increased lipofuscin density and darker “hypofluorescent” areas represent areas of decreased lipofuscin density [49,50].

One of the hallmarks of early and intermediate AMD is macular drusen [51], which form with RPE aging. Drusen are composed of lipofuscin containing dense lipids; carbohydrates; zinc; and proteins, including apolipoprotein B and E, as well as components of the complement system [52]. Recent grading systems, including the Age-Related Eye Disease Study (AREDS), and the Beckman Initiative for Macular Research Classification Committee have classified drusen based on drusen type and size to associate drusen regression with or without RPE atrophy in CNV or GA of late AMD [53,54,55].

A recent study classified drusen-associated atrophy stages based on FAF and histological findings in eyes with late AMD [56]. In stage 2, the earliest stage with detectable findings, FAF exhibited uniform hyperautofluorescence, indicating photoreceptor photopigment loss, whereas hypoautofluorescence in stages 3 and 4 corresponded to varying degrees of RPE atrophy. The FAF appearance is initially hyperfluorescent (stage 2), followed by a hypoautofluorescent center surrounded by hyperautofluorescent borders when associated with focal areas of RPE atrophy (stage 3), and hypoautofluorescent lesions with complete RPE loss (stage 4) [56]. As the disease progresses through stages, the proportion of lipid within the drusen decreases relative to the proportion of calcification, with 80% of the drusen being refractile at the advanced stage 4. The refractile drusen appear as yellowish-white, glistening lesions and are associated with an increased risk of developing late AMD; however, they are undetectable on FAF alone [57].

Of note, it was reported that cuticular drusen are strongly associated with late AMD [58]. Eyes with cuticular drusen can develop NV or acquired vitelliform lesions (AVL) [59], which may regress to GA or RPE atrophy [58]. In longitudinal studies, GA developed in 19.0% of eyes with cuticular drusen over a mean follow-up period of 40 ± 18 months, whereas GA developed in 28.4% and NV in 12.5% over a 5-year follow-up period [60]. The cuticular drusen apex is steep and is where the atrophic RPE is located. FAF is an effective method to detect cuticular drusen with the display of numerous hypoautofluorescences corresponding to the apex of the cuticle drusen with hyperautofluorescent rims. However, some FAF imaging cameras with different excitation wavelengths may not visualize these drusen [61].

Studies showed that RPD (also named subretinal drusenoid deposits) are highly associated with late AMD, such as GA, Type 3 macular NV, and drusenoid PED [62,63,64]. Soft drusen are located beneath the RPE whereas RPD are found on the surface of RPE [65]. Studies classify RPD into three types [66,67], in which the ribbon/reticular type is likely to progress to advanced AMD, including GA and Type 3 macular NV [68,69,70]. Like cuticular drusen, eyes with RPD can develop NV, or regress to GA or outer retina atrophy with focal photoreceptor loss and choroidal thinning [71]. FAF may demonstrate a reticular pattern in eyes with RPD; however, studies indicate that FAF is not the most specific method for detecting RPD [72].

Assessing the risk of late AMD depends on stratifying the types of drusenoid deposits and RPE abnormalities, and requires correctly evaluating imaging characteristics. The high-contrast differentiation of atrophic versus non-atrophic areas shown by FAF is a reliable image quantification of lesion area [73]. Currently, conventional blue light excitation with excitation wavelength of 488 nm is the most popularly used mode for FAF imaging. However, macular pigment blocks blue light, resulting in a relatively diminished signal intensity at the fovea, which appears as a zone of hypofluorescence [74]. Therefore, blue-light FAF may result in an overestimation of atrophic patch size and be mistaken for central atrophy involvement. The relative hypofluorescence of the fovea could mask an atrophic area, making it challenging to identify minimal central atrophic changes or adjacent paracentral atrophic margins [75]. The quality of the blue FAF signal may also be affected by pupil size or media opacity such as cataracts or vitreous opacity. FAF imaging systems include cSLO systems and flash fundus camera-based systems. FAF imaging with two excitation wavelengths (488 nm and 514 nm) is obtained via cSLO, while fundus camera autofluorescence relies on excitation wavelengths in the green to orange range (510–610 nm). One study reported that green-light FAF images (514 nm) are superior to blue autofluorescence (488 nm) for the evaluation of small central GA lesions [74]. Although the measurement of atrophic lesion size in current clinical studies depends mainly on blue-light FAF, green-light FAF appears to be a more accurate, and a potentially important evaluation tool for central MA progression in future studies. 

In certain phenotypic variants of GA, the loss of contrast between intact and atrophic RPE can have an altered FAF appearance, which differs from the markedly hypoautofluorescent images in other forms of GA [76]. In eyes with hemorrhagic nAMD or late nAMD with MA, the FAF signal may be reduced, and it is difficult to distinguish between atrophy and areas of fibrosis using FAF alone [67]. Recently, blue-light FAF has been utilized in conjunction with near-infrared reflectance (NIR), which is unaffected by luteal pigment and enhances foveal evaluation. NIR is characterized by a long excitation wavelength (820 nm diode laser) [77], that avoids the absorption of a shorter wavelength of light (480 nm) by melanin and lipofuscin granules at the RPE level, thereby allowing visualization of the retina and choroid [78,79]. Specifically, NIR reveals sub-RPE lesions effectively. Refractile drusen, for instance, are highly reflective. They are seen as glistening dots using NIR, but are undetectable using FAF [80] (Figure 1E). Studies have reported that NIR has a very high sensitivity for detecting RPD [30,72,81,82]. However, systematic validation studies for NIR alone in the detection of atrophic AMD are still lacking. Hence, FAF combined with other diagnostic modalities such as NIR may improve visibility of the obscured atrophic demarcated areas compared to using FAF alone. Furthermore, widefield imaging devices can be used for the acquisition of FAF and both FA and ICG-A. 

### 3.4. Optical Coherence Tomography

A noninvasive imaging modality, OCT, utilizes transversely scanned short coherence length light with interferometry to generate 2-dimensional and 3-dimensional cross-sectional maps of the retina and choroid with micrometer-level resolution [83]. While FAF is valuable for quantifying RPE loss in MA, it does not discern non-RPE layer changes [75]. The classic definition of atrophy has been revised to incorporate changes in the outer retinal layers based on OCT findings [76]. A classification system and criteria for OCT-defined atrophy associated with AMD has been proposed by the International Classification of Atrophy Meetings (CAM). According to the CAM study group, the OCT finding of atrophy undergoes an evolution of four different stages [84]: (1) incomplete outer retinal atrophy; (2) complete outer retinal atrophy; (3) incomplete RPE and outer retinal atrophy (iRORA) (Figure 1F,G); and (4) complete RPE and outer retinal atrophy (cRORA). Of note is that these terms apply to atrophy in both non-neovascular (dry) and neovascular (wet) forms of AMD [76]. The correlation between FAF changes and the four distinct atrophy categories is currently unknown. The correlation between hypoautofluorescence in FAF and the category of OCT-defined atrophy requires further investigation.

It is crucial that high resolution 3-dimensional OCT help identify the early phase of the atrophic process prior to lesion detection in 2-dimensional FAF [76,77,78,79,80]. The high axial resolution of Fourier-domain OCT devices, including spectral-domain OCT (SD-OCT) and swept-source OCT (SS-OCT), allows for the study of atrophy to quantify specific retinal layer loss. The wide application of SD-OCT has revolutionized the diagnosis and management of nAMD as it can provide assessment of risk and treatment prognosis, including the need for repeated anti-VEGF injections and other therapeutic interventions [85]. OCT has evolved into an effective imaging modality for evaluating early AMD changes. High-resolution OCT detects the presence of drusen and pigmentary changes in the early stages of AMD [53,86,87], but SD-OCT provides important information regarding changes in retinal layers such as the outer plexiform layer (OPL); inner nuclear layer (INL); external limiting membrane (ELM); and ellipsoid zone (EZ). Unlike previously reported non-unique risk factors for the development of atrophy, such as hyperreflective foci and particular drusen characteristics (including heterogeneous internal reflectivity, and maximum drusen height and choroidal thickness beneath the drusen) [88,89], SD-OCT may detect unique early features such as the subsidence of the OPL and INL, and a hyporeflective wedge-shaped band within the limits of the OPL, that are present prior to development of drusen-associated atrophy and represent significant risk and [90]. In addition, SD-OCT can detect early morphological changes before conventional diagnostic instruments. For instance, in one study, SD-OCT showed that drusen-associated atrophy was already present in 2.9% of patients’ eyes classified as having intermediate AMD using color fundus photography [55,90]. In another study, the pathological SD-OCT features occurred approximately one year prior to the development of definitive drusen-associated atrophy [90]. This may enable treatment to halt the progression of atrophy to be considered at an earlier time point [91,92,93], before late atrophic changes are detectable via conventional diagnostic methods. 

A consensus was reached on the descriptions of imaging characteristics associated with atrophy or atrophy progression risk in eyes with AMD [84]. OCT features associated with risk for atrophy include intraretinal hyperreflective foci; extracellular deposits (soft drusen, drusen with hyporeflective cores, cuticular drusen, drusenoid PED, and subretinal drusenoid deposits); hyperreflective crystalline deposits in the sub-RPE basal lamina (BL) space; and acquired vitelliform lesions [81,88,89,94,95,96,97,98]. As drusen regress, the overlying retinal layers progress to atrophy that can be detected by OCT imaging. Outer retinal atrophy features included INL and OPL subsidence; ELM descent; a hyporeflective wedge-shaped band within the Henle fiber layer, often accompanied by RPE disturbance and increased signal hypertransmission into the choroid; and ELM and EZ disruption [90,99,100,101]. For iRORA to be present, three OCT features, including photoreceptor degeneration; RPE attenuation or disruption; and increased signal transmission into the choroid are required [102]. However, a minimum size limit for iRORA was not proposed. The study further reported that iRORA will progress and develop into cRORA over a variable time period ranging from months to years [102]. A model was then developed to estimate potential future atrophy growth regions and identify predictive biomarkers. The most predictive SD-OCT biomarkers were thickness loss of bands; reflectivity of bands; thickness of RPD; GA projection image; increased minimum retinal intensity map; and GA eccentricity, based on quantitative characteristics of GA [103]. SD-OCT can ultimately detect presence of fluid accumulation within and beneath the retina, as observed in CNV cases.

The anatomical correlations of the individual bands identified utilizing an SD-OCT line scan are well established [104]. The distance interval between scans must be small enough to avoid missing pathologic characteristics such as drusen, RPD, and pigment migration into the inner retina. Scanning with a spacing of 125–250 μm is suggested for the detection of RPD. This can indicate rapid atrophy progression and the volume rendering of outer retinal tubulations [105,106]. However, a less dense scan is typically preferred in longitudinal, large-scale clinical trials as a trade-off to achieve a shorter acquisition time [30]. In contrast to SD-OCT, which employs an interference spectrum acquired through spectral splitting and a low-coherence light source, SS-OCT typically utilizes a broad band sweep source in which the wavelength of the light source fluctuates over time [107]. The longer wavelengths of SS-OCT enable better penetration into the choroid, and excellent reproducibility and repeatability of choroidal thickness measurements [108,109]. Using SS-OCT imaging to detect hypertransmission into the choroid, cRORA [84] and iRORA [102] can be identified via an en face slab with boundaries beneath the RPE.

Recent polarization-sensitive optical coherence tomography (PS-OCT) was utilized to estimate the melanin content of RPE, with the surface of the atrophy area having high entropy (low melanin) [110]. In contrast, standard imaging tools, such as CPF or FAF, are incapable of determining the degree of RPE pigmentation. Entropy values in PS-OCT are suggested as a detective tool to assess the degree of RPE pigmentation and, hence, the health of RPE; however, further research is needed to validate this. OCT imaging system combined with laser doppler flowmetry = optical doppler tomography (ODT), allows the quantitative imaging of fluid flow in a highly scattering medium [111]. In addition to traditional structural OCT, ODT evaluates tissue function in AMD eyes with possible abnormal ocular circulation. However, clinical applications of this functional OCT are still in their early stage of development. Similarly, an angiogram without fluorescent dye injection, phase contrast optical coherence tomography (PC-OCT), was developed to image retinal microvasculature [112]. In the atrophic area of GA, a PC-OCT angiographic image showed large choroidal vessels and loss of the overlying superficial choriocapillaris. However, there is still no consensus whether patients with GA truly have an absence of flow or only a reduction in flow [113]. Hence, these findings warrant further clarification.

### 3.5. Optical Coherence Tomography Angiography

Imaging capable of providing appropriate visibility of the choriocapillaris and choroid has improved our understanding of atrophic and nAMD. While FA allows visualization of the retinal vasculature but not the choriocapillaris, ICG-A has not been widely utilized for choriocapillaris visualization in AMD due to its lack of depth resolution and inability to differentiate between choriocapillary blood flow and that of deeper choroidal vasculature [114,115,116]. In contrast, OCTA allows depth-resolved imaging of the retinal, choriocapillarial, and choroidal vasculatures. OCTA generates three-dimensional images of vasculature without dye injection. Repeated imaging of stationary tissue with OCTA produces a series of identical B-scans; when there is motion due to blood flow, the repeated B-scans will alter, and the changes can be quantified [117,118,119,120]. Recent OCTA studies demonstrated choriocapillaris loss across a spectrum of AMD phenotypes, including soft drusen, RPD [121,122,123,124], and CNV [125]. OCTA also allows for the evaluation of choroidal layers within and around atrophic lesions (Figure 1H). Some studies found that the area surrounding the GA margin has greater choriocapillaris flow loss than the area of RPE atrophy or GA [126], indicating that choriocapillaris degeneration may occur prior to the development of GA and may be a prognostic factor for atrophic progression [127,128,129,130]. However, there are conflicting findings that choriocapillaris loss was linearly related to or less than RPE loss in GA [131], leading to the conclusion that the RPE appeared to be the primary target in GA [132,133]. In the GA region, it may be difficult to distinguish choriocapillaris flow impairment from atrophy due to OCTA’s lower limitation in detecting slow blood flow. Increasing the interscan time can increase the sensitivity of OCTA to slow flows, but it also increases eye motion artifact noise [134]. Hence, both the sensitivity to slow flow and the potential artifacts must be considered when interpreting OCTA data [131]. In addition, OCTA limitations include acquisition time and field when used with conventional OCT. Therefore, dense, high-quality SS-OCT scans are required to obtain reliable OCTA results. The Consensus on Atrophy (CAM) study group recommended OCTA may be optionally included in studies on non-neovascular and neovascular AMD for exploratory purposes [135].

## 4. Current Approach and Future Directions

In this review, we attempt to correlate relevant diagnostic tools to corresponding features of AMD and findings that emerge prior to the formation of MA, based on consensus definitions.

### 4.1. Imaging Algorithm

For AMD-affected eyes with no clinical indication of active or regressed CNV; a baseline CFP; or UWF imaging, OCT, FAF, and FA is recommended. Throughout the follow-up, CFP, FAF, OCT can be performed at regular intervals to monitor progression. Angiography (FA, ICG-A and OCTA) may be indicated in the event that a neovascular process is suspected in follow-up. For patients with macular atrophy from nAMD, CFP (Figure 2A); UWF (Figure 2B); FA (Figure 2C); FAF; SD-OCT; and OCTA is recommended at baseline and selected follow-up visits (1–3 months) [136]. ICGA (Figure 2D) would be considered in differentiation of RAP, PCV, or CSC [137].

For optimal detection and measurement of late AMD atrophy and associated characteristics of MA, multimodal imaging is advised in clinical settings. Imaging includes CFP; FAF (Figure 2E); and confocal NIR (Figure 2F), and high-resolution OCT volume scans (Figure 2G) or OCTA (Figure 2H) should be acquired at regular intervals throughout the study to detect, quantify, and monitor progression of atrophy. The most important differential diagnoses of atrophic late AMD are the gene ABCA4, PRPH2, and BEST1 inherited macular dystrophies [138], including Stargardt disease (STGD1), Best disease (BD), or pseudo-Stargardt multifocal pattern dystrophy (PSPD) in late atrophic stages [139]. The atrophic lesions in inherited macular dystrophies makes diagnosis challenging when relying on similar fundus appearance. In contrast, machine learning algorithms can serve as a very useful tool in the automated quantification of pathologic characteristics [85]. A study indicated that a deep learning model to classify atrophy on FAF imaging can accurately differentiate atrophy caused by GA from that caused by inherited macular dystrophies [140]. Recently, machine learning (specifically deep learning techniques), has been applied to FAF and OCT imaging to detect and classify GA [140,141,142,143]. More recently, a deep learning model using convolutional neural networks was developed and validated for segmentation of the 13 most common features related to early and late nAMD [144]. In addition, the application of deep learning algorithms to the prediction of progression to nAMD has favorable results [145]. The utilization of artificial intelligence, such as the deep learning model in automated OCT analysis, could potentially facilitate the early detection and predict progression of AMD [146], thereby expanding the future therapeutic window of opportunity. As the deep learning model further develops, it may be able to predict the risk of developing MA, offer early detection and provide customized treatment strategies.

### 4.2. New Modalities for Imaging MA-Microperimetry, Adaptive Optics, Home-Based OCT

Microperimetry is an automated perimetry system with a non-invasive technique to spatially map retinal sensitivity. Reductions in retinal sensitivity occur rapidly and precede visual acuity changes in AMD [147,148,149]. Microperimetry has eye tracking capability that measures differential light sensitivity (DLS), which is the minimum luminance of a white-spot stimulus that can be perceived on a white superimposed background of uniform luminance. Mean sensitivity (MS) quantifies the average DLS across all stimulus locations. Localized decreases in retinal sensitivity have been reported in GA precursor lesions [150], in which the deterioration of visual function can occur months to years before the patient experiences visual problems [148]. Microperimetric sensitivity has also been associated with drusen volume, RPD, and extent of pigmentary changes [151]. In eyes affected by GA, microperimetry detects an increasing number of scotomatous points and a decline in MS over time, which correlates anatomically with an increase in atrophic size [152] and a reduction of the inner segment–outer segment junctional layer of photoreceptors suggesting disease progression [153,154]. Eyes with atrophic AMD were found to have decreased sensitivity in all retinal regions, including those at the GA margin or outside atrophic lesions, indicating that patients with GA have a more extensive functional deficit than those with mild/intermediate AMD [155]. Currently, anatomic assessment of AMD via multimodal fundus imaging is commonly used to diagnose and monitor the disease; however, microperimetry can identify dysfunction in patients with AMD and quantify late-stage progression by measuring local functional deficits in the retina.

Adaptive optics (AO) imaging systems improve resolution compared to traditional retinal imaging by eradicating the strong signal from photoreceptor outer segments and the highly scattered nerve fibers and vessels [156]. Recently, AO has been incorporated in flood illumination fundus cameras, cSLO and OCT [157,158] to detect weakly reflective structures such as photoreceptor inner segments, RPE and retinal ganglion cells [159,160]. In late AMD eyes, the boundaries of atrophic areas may appear clearly defined or ill-defined on AO imaging. The hyporeflective nature of distinct borders indicates the existence of melanin clusters. Melanin redistribution is pronounced in late AMD [161,162], with the presence of hyporeflective clusters (HRCs) that are presumed to be detached RPE cells [161], microglia [163], or monocyte macrophages that have phagocytized RPE cells [164]. HRCs are dispersed within and around atrophic regions of the RPE layer. Additionally, time-lapse photography demonstrated the dynamic nature of the HRC redistribution. HRCs were identified during their developing stages, indicating that they developed concurrently with, or even before, the start of MA [165]. When monitoring AMD patients, comparing subsequent AO images can demonstrate MA progression.

Home-based OCT monitoring allows disease monitoring from the comfort of patients’ homes and has emerged as a tool for the timely detection and management of late-stage AMD. The feasibility of home-based OCT monitoring with self-scanning, coupled with image analysis software for detecting fluid volume in nAMD, was shown to be highly correlated with manual human grading [166]. This provides clinicians with valuable data to monitor treatment strategies and reduces the burden of disease follow-up for patients, their caregivers, and providers. However, challenges such as patient adherence, data interpretation, and the need for clinician oversight remain significant hurdles. Despite these challenges, the integration of home-based OCT monitoring holds promise in enhancing the overall management of nAMD and reducing treatment burden on healthcare systems. Interestingly, home-based OCT is currently only being studied to monitor eyes with nAMD; monitoring non-nAMD fellow eyes may also be beneficial.

With increasing life expectancy and an aging global population, late AMD poses a considerable and expanding threat to society, and the resulting visual impairment of MA represents an enormous resource burden. To identify the optimal combination of imaging modalities, it will be crucial to conduct validation studies accordingly. As novel imaging technologies emerge in the future, it will be essential to revise these recommendations consequently. The visual outcomes of patients with nAMD have improved due to the development of anti-VEGF medication [167,168,169,170]. Novel therapeutic techniques, such as gene therapy using recombinant adeno-associated virus vectors delivering VEGF-inhibitory compounds, and subsequent expression for long-term nAMD control, are emerging [171,172]. However, a staggering percentage of nAMD patients stabilized via anti-VEGF treatment still go on to develop MA: more than 98% at 7 years in some studies [173]. New treatments for GA in dry AMD include complement pathway C3 and C5 inhibitors [174,175,176,177]. In all of these efforts, the ability to utilize advances in imaging modalities to detect and document early disease findings remains a critical stepping-stone to future therapies.

## Figures and Tables

**Figure 1 diagnostics-13-03635-f001:**
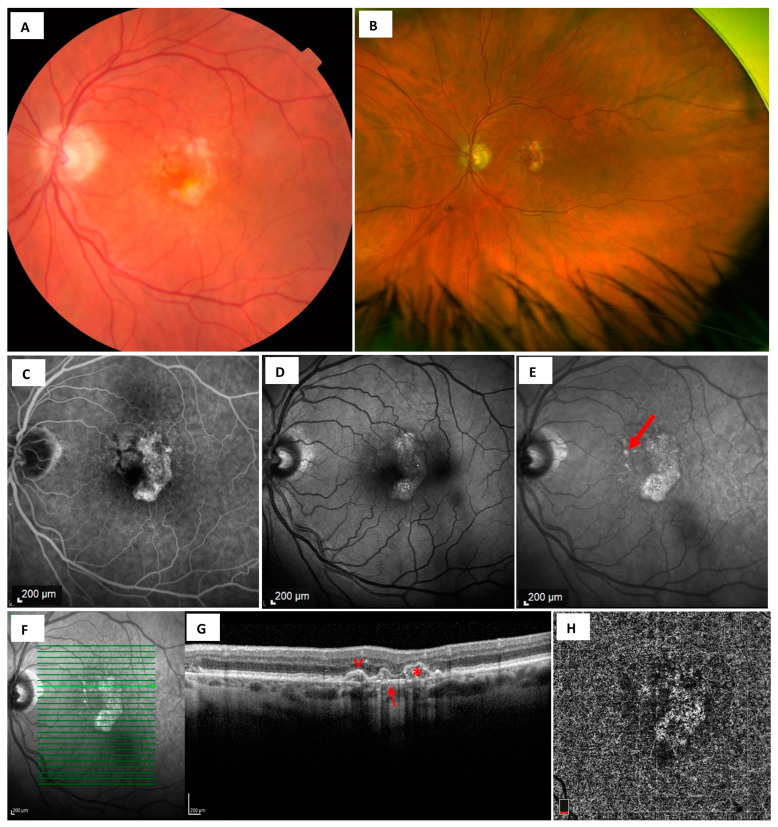
Images of geographic atrophy (GA) with dry age-related macular degeneration (AMD). (**A**) Color fundus photography (CFP) shows multifocal retinal atrophy, and drusen without clear delineation of atrophic lesion boundaries. (**B**) Ultra-widefield scanning laser ophthalmoscopy shows GA in the presence of dry AMD. (**C**) Atrophic patches in fluorescein angiography (FA) appear as well-demarcated, hyperfluorescent areas due to enhanced visualization of choroidal fluorescence caused by the loss of RPE cells (window defect), which would normally diminish the transmission of fluorescein fluorescence. (**D**) Fundus autofluorescence (FAF) shows focal areas of hypoautofluorescence, indicating photoreceptor photopigment loss, and hyperautofluorescence, corresponding to varying degrees of RPE atrophy. (**E**) Near-infrared reflectance (NIR) image with relatively nonspecific reflectivity. NIR has a higher sensitivity for detecting refractile drusen as glistening dots (arrow) that are undetectable using FAF. (**F**) Scan position as indicated by bold green arrow line in SD OCT corresponding to (**G**) incomplete RPE and outer retinal atrophy (iRORA) at fovea, with hypertransmission under a region of absent RPE and outer retinal bands, as well as an undiscerned outer nuclear layer (arrow). Parafoveal region shows an elevated drusenoid pigment epithelial detachment (PED) (asterisk). The retinal pigment epithelium (RPE) with a focal thickening at the apex (reticular pseudodrusen, RPD), corresponding to hypertransmission in the choroid (arrowhead). RPD, as a major risk factor for progression to late AMD, is overlooked with standard CFP. (**H**) Optical coherence tomography angiography (OCTA) with reduced choriocapillaris identified via an en face slab.

**Figure 2 diagnostics-13-03635-f002:**
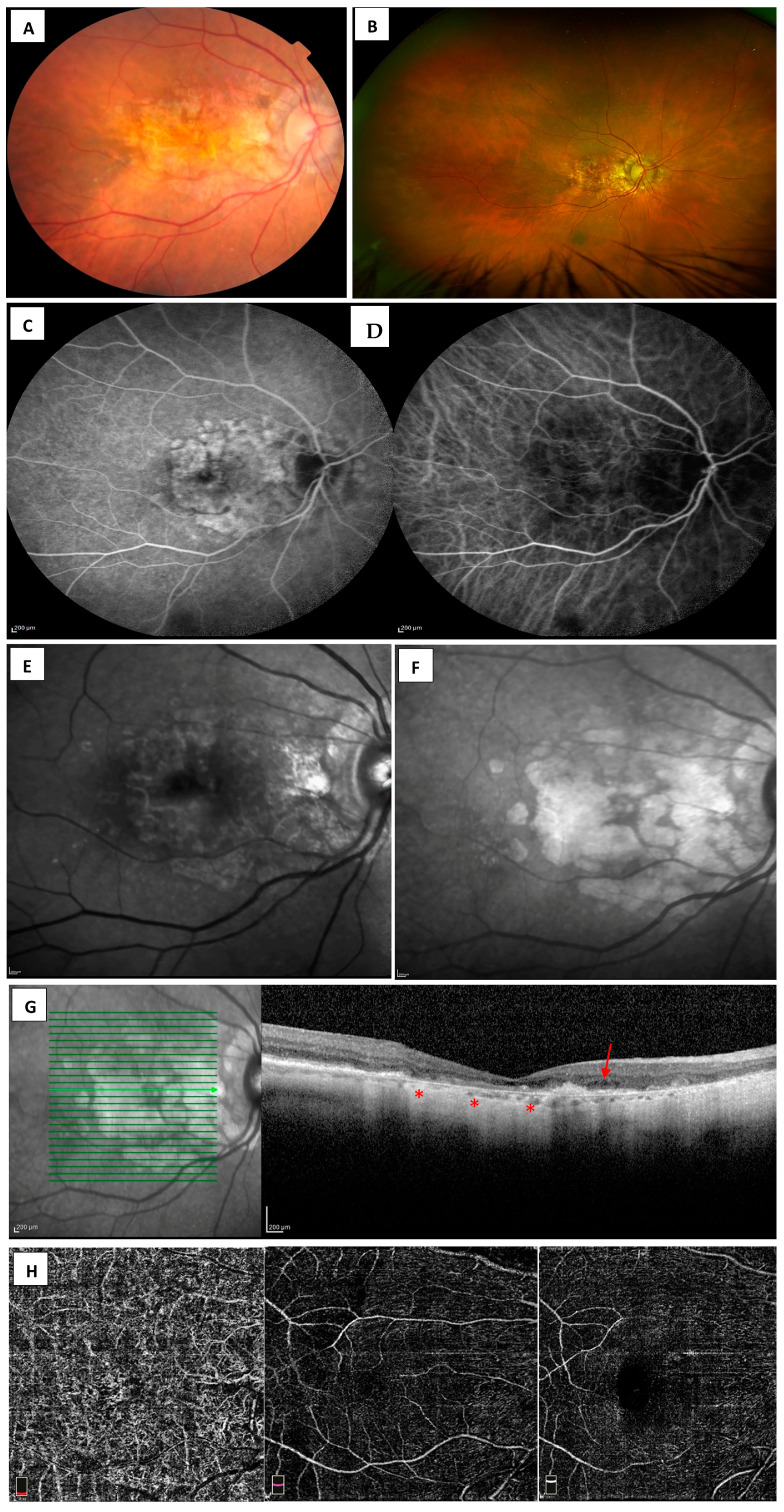
Images of macular atrophy (MA) with neovascular age-related macular degeneration (nAMD). (**A**) Color fundus photography (CFP) shows large area of coalesced retinal atrophy with sparse areas of intact RPE without delineation of atrophic lesion boundaries. (**B**) Ultra-widefield scanning laser ophthalmoscopy shows MA in the presence of nAMD. (**C**) Atrophic patches in fluorescein angiography (FA) appear as demarcated hyperfluorescent areas, whereas (**D**) in indocyanine green angiography (ICG-A), they appear as minimally discernable hypofluorescent areas. (**E**) Fundus autofluorescence (FAF) shows a demarcated region hypoautofluorescence surrounded by a rim of weak hyperautofluorescence. (**F**) Atrophic patches, including some that are undetectable usingFAF, are discernable using near-infrared reflectance (NIR). (**G**) Spectral-domain optical coherence tomography (SD-OCT) scan position as indicated by bold green arrow line with corresponding scan in the (**right panel)** that shows disrupted inner and outer retinal bands with region of complete retinal pigment epithelium atrophy (cRORA) (asterisk) with intraretinal fluid (arrow). (**H**) Optical coherence tomography angiography (OCTA) with reduced choriocapillaris (**right panel**), deep (**middle panel**), and superficial retinal plexus (**left panel**) identified via an en face slab.

**Table 1 diagnostics-13-03635-t001:** Retinal imaging techniques.

Techniques	Pros	Cons
Color fundus photography (CFP)	Conventional historical	Low contrast
	Closely resembles clinical ophthalmoscopy	Experienced photographers and patient cooperation needed
	Documents a wide range of fundus abnormalities and AMD-associated phenotypic changes, particularly hemorrhages and focal pigmentary changes.	Difficulties in precisely delineating lesion boundaries/quantification of atrophic size
		Sensitive to optical media
Multicolor imaging	High contrast	Less distinct structures due to chromatic aberration
	Reveals details of different retinal layers	Poor distinction between hemorrhages and pigmentary lesions
Ultra-widefield (UWF)	Visualization of wide field of the retina	Developing tool
	Peripheral abnormalities visible	Limited data available
		Unknown clinical significance of peripheral lesions in AMD
Fundus fluorescein angiography (FA)	Visualization of retinal vasculature	Invasive procedures
	Gold standard for NV detection and quantification	Limited imaging window after injection
	Excludes presence of concurrent NV	Risk of life-threatening allergic reaction
	Sharper contrast between atrophic and surrounding non-atrophic areas.	Other lesions or leakage may obscure boundary demarcation of atrophy
		No visualization of choriocapillaris
Indocyanine green angiography (ICG-A)	Visualization of choroidal vasculature	Invasive procedures
	Choroidal imaging for differential diagnosis of PCV, RAP, CSR, Stargardt disease and nAMD	Limited imaging window after injection
		Risk of life-threatening allergic reaction
		Deep choroidal vessels may interfere with outline of area of atrophy
Fundus autofluorescence (FAF)	Non-invasive	Sensitive to optical media
	High contrast, good atrophic lesion boundary discrimination	Refractile (calcified) drusen at advanced stage are undetectable
	Quantification of RPE loss	Overestimates size of atrophic patch at macula
		Mask an atrophic area due to relative hypofluorescence of the fovea
		Unable to discern non-RPE layer features
Near-infrared reflectance (NIR)	Non-invasive	No validation studies in the detection of late AMD
	Unaffected by luteal pigment in foveal evaluation	Complements other imaging techniques
	High sensitivity for reticular pseudodrusen	
	Not affected by optical media	
Spectral-domain OCT (SD-OCT)/Swept-source OCT (SS-OCT)	Non-invasive	Scan field depends on optics used in the system
	Three-dimensional	
	Detects morphology of retina layers, RPE, and choroid	
	Detects early AMD features	
Polarization-sensitive optical coherence tomography (PS-OCT)	Novel technique	No validation studies
	Assess RPE pigmentation	
Optical Doppler tomography (ODT)	Quantitative imaging of vasculature	Still limited data available
	Functional detection	
Phase contrast optical coherence tomography (PC-OCT)	Retinal microvasculature imaging	No consensus yet
Optical coherence tomography angiography (OCTA)	Non-invasive, no dye injection	Acquisition time and field when used with conventional OCT
	Three-dimensional images of vasculature	
	Evaluation of choroidal layers	
	Lower limitation in detecting slow blood flow	
	Administered at any time	

Abbreviations: AMD: age-related macular degeneration; CSR: chronic central serous chorioretinopathy; nAMD: neovascular AMD; PCV: polypoidal choroidal vasculopathy; RAP: retinal angiomatous proliferation. RPE: Retinal pigment epithelium.

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
