# Peer review of "Recent Advances in Imaging Macular Atrophy for Late-Stage Age-Related Macular Degeneration"

_diagnostics, 2023, doi:10.3390/diagnostics13243635_

Round 1

Reviewer 1 Report

Comments and Suggestions for Authors

The authors attempt to provide a review of the literature on imaging modalities and features of late AMD. The topic is important, however the execution is somewhat lacking. I have the following comments:

Major points:

- The review's main purpose is to review imaging of late AMD, however, the authors go on deep discussions of prevalence, stages, therapeutics of AMD, as well as risk factors for AMD.

- The introduction is too long and needs to be shortened to only discuss a brief overview of 1) what constitutes "late AMD", and 2) Why is it important to image late AMD (especially with the new therapeutics on the horizon, to select patients for trials or newly available therapies).

- I recommend the authors organize the review into two sections. According to the AREDS classification, late AMD is either GA or neovascular AMD, and the authors should discuss each individually.

- I recommend they then discuss under each section the following subheadings: clinical presentation, fundus photography, FAF, dye-based angiography, OCT, and OCTA.

- Fundus photography should be discussed in terms of regular color photos, multicolor photos, ultrawidefield machines..etc

- Neovascular AMD is not properly discussed. The authors should discuss how to recognize nAMD using the aforementioned subheadings and contrast imaging techniques. The authors should also discuss the different "pathways of fluid leakage in AMD" and the importance of not jumping to anti-VEGF whenever fluid is seen, which is summarized in a recent retina editorial.

- The review lacks images, which is crucial for an article discussing imaging modalities. The authors should add the appearance of typical cases of GA and nAMD on color photos, FAF, FA, ICGA, OCT, and OCTA.

- The differential diagnosis of eyes with late AMD, especially GA, is recommended to be added.

- A summarizing algorithm or recommendation for imaging eyes with AMD is recommended to be added, based on the best available evidence to date.

Minor points:

- Remove discussion of relation between anti-VEGF and MA as it is irrelevant to the review's subject.

- Remove discussion of drusen type, risk, and imaging appearance as the review is set to only deal with "late AMD" of which drusen are not a part of.

- Lines 319-321: EDI OCT is not a new technology and has been around for about a decade now.

- Section about assessment of MA using visual function is not scientifically sound and I recommend its removal.

- No need to mention details of how dye angiography works as this is common knowledge to the readership of the article.

Comments on the Quality of English Language

Moderate English editing for punctuation, grammar, and spelling. For example, sentences should not start with acronyms (e.g. "nAMD" or "MA"). 

Author Response

We thank you and reviewer for the valuable comments. We have revised the manuscript accordingly and provided the following point-to-point responses:
(Please see the attachment)

Reviewer 1 Comments:

Comment #1:  The introduction is too long and needs to be shortened to only discuss a brief overview of 1) what constitutes "late AMD", and 2) Why is it important to image late AMD (especially with the new therapeutics on the horizon, to select patients for trials or newly available therapies).

Reply: we revised the introduction section and discussed overview of

  • what constitutes "late AMD" in paragraph 1, line 82-91: “ In the late stage of AMD, significant visual impairment is due to dry AMD with geographic atrophy (GA) or wet AMD with choroidal neovascularization (CNV). In GA, in the setting of dry AMD, the retinal pigment epithelium, choriocapillaris, and photoreceptors are progressively atrophied[7] . While in wet neovasular AMD (nAMD), choroidal neovascularization penetrates Bruch’s membrane, leading to the leakage of fluid, lipid, blood and resulting in retinal fibrosis. Macular atrophy (MA), an anatomic endpoint of AMD representing both GA in dry AMD and CNV in nAMD, is characterized by the permanent degradation of the retinal pigment epithelium (RPE) and overlying photoreceptors. Multiple studies have shown that natural progression to MA is the final common pathway in both nAMD and dry AMD disease progression” constitutes "late AMD", and
  • the importance of imaging late AMD in paragraph 3, line 109-114 “Currently, there are no standard and effective treatments for MA despite emerging innovative therapies. With new therapeutics on the horizon, choosing patients for trials or newly developed therapies that will improve the clinical course of AMD and starting early detection may help delay or halt disease progression. Hence, it is important to detect incident MA at the first appearance to identify subsequent progression to central macular involvement over time.”

Comment #2:  I recommend the authors organize the review into two sections. According to the AREDS classification, late AMD is either GA or neovascular AMD, and the authors should discuss each individually.

Reply: We understand and agree that late AMD is either GA or neovascular AMD. The term “GA” has been used with various definitions in the past. In the current article, we use the term “MA” for complete retinal pigment epithelium (RPE) and outer retinal atrophy, as multiple studies have shown that natural progression to MA is the final common pathway in both nAMD and dry AMD disease progression (introduction paragraph 1, line 90-91). In this sense, instead of discussing each individually, we described atrophic features in MA that can be present in eyes with no evident signs of active or regressed NV (dry AMD) and in eyes with active or regressed NV outside the NV region (nAMD).

The current review article has already reached word limit of 5093. Due to word limitations, we are unable to discuss each in detail; however, we have discussed neovascular AMD: in Section 3.2 Dye-based Angiography, we do describe “neovascularization in nAMD, may also result in an increased fluorescence signal or progressive dye leakage and therefore obscure the boundary demarcation of atrophy.” , Section 3.3. Fundus Autofluorescence, we describe “In eyes with hemorrhagic nAMD or late nAMD with MA, the FAF signal may be reduced”, and in Section 4. Ongoing trends in management and research, we discuss the “Differential diagnosis of nAMD”.

Comment #3:   I recommend they then discuss under each section the following subheadings: clinical presentation, fundus photography, FAF, dye-based angiography, OCT, and OCTA.

Reply: We revised in Section 3 per suggestion.

Comment #4:  Fundus photography should be discussed in terms of regular color photos, multicolor photos, ultrawidefield machines..etc

Reply: We revised in Section 3.1. per suggestion.

Comment #5:  Neovascular AMD is not properly discussed. The authors should discuss how to recognize nAMD using the aforementioned subheadings and contrast imaging techniques. The authors should also discuss the different "pathways of fluid leakage in AMD" and the importance of not jumping to anti-VEGF whenever fluid is seen, which is summarized in a recent retina editorial.

Reply: We thank you for the comment, but as addressed in reply#2, we discussed the final common pathway (MA) features in both nAMD and dry AMD disease. Due to word limitations, we may discuss Neovascular AMD as the separate subject of another article.

Comment #6:  The review lacks images, which is crucial for an article discussing imaging modalities. The authors should add the appearance of typical cases of GA and nAMD on color photos, FAF, FA, ICGA, OCT, and OCTA.

Reply: We added figures per suggestion.

Comment #7:  The differential diagnosis of eyes with late AMD, especially GA, is recommended to be added.

Reply: We added differential diagnosis of eyes with late AMD, especially GA, in Section 4, line 440 – 449:” The most important differential diagnoses of atrophic late AMD are the gene ABCA4, PRPH2, and BEST1 inherited macular dystrophies, including Stargardt disease (STGD1), Best disease (BD), or pseudo-Stargardt multifocal pattern dystrophy (PSPD) in late atrophic stages. The atrophic lesion in inherited macular dystrophies makes diagnosis challenging when rely on similar fundus appearance. In contrast, machine learning algorithms can serve as a very useful tool in the automated quantification of pathologic characteristics. A study indicated that a deep learning model to classify atrophy on FAF imaging can accurately differentiate atrophy caused by GA from its masqueraders - inherited macular dystrophies. Recently, machine learning, specifically deep learning techniques have been applied to FAF and OCT imaging to detect and classify GA”

Comment #8:   A summarizing algorithm or recommendation for imaging eyes with AMD is recommended to be added, based on the best available evidence to date.

Reply: We added a summarizing algorithm in Section 4, line 458-466:” For AMD eyes with no clinical indication of active or regressed CNV, a baseline CFP is recommended, while an ICG-A test may be used in some instances. Throughout the follow-up, FA may be required at any visit to assess for NV. For patients with nAMD, FA is recommended at baseline and selected follow-up visits while an ICG-A can also be used for visualization of the vasculature. For optimal detection and measurement of late AMD atrophy and associated characteristics of MA, multimodal imaging is advised in clinical settings. Imaging include CFP, FAF, confocal NIR, and high-resolution OCT volume scans images or OCTA should be acquired at regular intervals throughout the study to detect, quantify, and monitor progression of atrophy.”

Comment #9:   Remove discussion of relation between anti-VEGF and MA as it is irrelevant to the review's subject.

Reply: we removed the discussion per suggestion.

Comment #10:    Remove discussion of drusen type, risk, and imaging appearance as the review is set to only deal with "late AMD" of which drusen are not a part of.

Reply: We thank you for the comment; we removed some parts of the discussion. However, we still described the imaging findings, as this is supported in Guymer 2023 Lancet, which has revealed the importance of certain types of drusen, such as reticular pseudodrusen (RPD), cuticular drusen, as distinct AMD phenotypes that are associated with an increased risk of GA in late AMD.

~Guymer et al. Lancet. 2023 Apr 29;401(10386):1459-1472.

Comment #11:  Lines 319-321: EDI OCT is not a new technology and has been around for about a decade now.

Reply: we deleted per suggestion.

Comment #12:  Section about assessment of MA using visual function is not scientifically sound and I recommend its removal.

Reply: we deleted per suggestion but briefly described in Section 2.1 as some studies reported that the assessment using visual function may quantify late-stage progression.

Comment #13:  No need to mention details of how dye angiography works as this is common knowledge to the readership of the article.

Reply: we removed the detail per suggestion.

Reviewer 2 Report

Comments and Suggestions for Authors

From a reviewer’s point of view, overall, the authors have done a clear review study of Age-related macular degeneration (AMD) diagnostic by imaging methods that are widely used. The introduction is well-written and explains extensively about AMD. The authors have provided a detailed explanation of the different imaging techniques used for AMD diagnosis.

Comments:

1.    Authors should consider adding a section for current widely used methods used for AMD diagnosis in clinical settings. Methods like dilated eye examination, Visual Acuity test, Amsler grid test, fundus photography, OCT, Fluorescein angiography, Electroretinogram, etc., and other methods should be discussed shortly prior to going for the Assessment using the imaging methods section. Even though authors have discussed them in other sections like ERG in section 3, and OCT in section 2, it is advised to start off with a brief overview of these types of diagnostic methods, which are currently widely used in clinical settings.

2.      Authors must consider using images of all discussed AMD imaging techniques. This can be done by including images from recently published articles on individual techniques for AMD diagnosis and reproducing them in their review article after getting the necessary permissions from respective publishers.

3.   Line 251: Abbreviate OCT. Acronyms should be abbreviated at their first.

4.      Can the reviewers provide a table comparing the pros and cons of all techniques that have been explained in their review article?

5.    Comparatively show the resolution, non-invasiveness, and accuracy/reliability of each of the discussed diagnostic methods. This can be in diagrammatic (preferably) or in table format.

6.   Can the authors give a short note on the effective utilization of these techniques being relied on in clinical scenarios? Adding this in the end section will be of more useful for readers

7.     It is true that among the different versions of OCT, OCTA is a go-to when assessing for eye imaging. There have been several research with different types of OCT like PS-OCT, Phase OCT, ODT, and other variations useful in the diagnosis of AMD. Authors should address these variations and their usefulness in different scenarios.

8.      Lines: 319 to 320, this is misleading and wrong, it is not like SS-OCT offers a larger scan area. This depends on the optics used in the system. Besides, SS-OCT may be introduced later on, but it is not a new technique. It is a variation that has been well-reported.

9.    Enhanced Depth in OCT can be achieved mu multiple methods using variations and different optical setups. Reporting it as one specific variation of OCT can be misleading to readers.

10.     In section 4, can the authors include more in details of the different diagnostic methods, future trends, and expected advancements that can greatly help doctors to better identify and early diagnosis of AMD?

11.      Also, under section 4, can the authors include the use of Deep-learning and other Machine learning methods that are being currently explored and reported by researchers and doctors alike for a better understanding of different imaging diagnostics and for early diagnostics? Also, even a short mention of how AI-assisted image analysis and machine learning can be expected to further advance understanding and early diagnosis of AMD. 

Comments on the Quality of English Language

The article is well written, but the flow of the article can be improved, and it needs minor to moderate changes with the English Language. Given the authors address all comments, the manuscript can be considered for acceptance.

Author Response

We thank you and reviewer for the valuable comments. We have revised the manuscript accordingly and provided the following point-to-point responses:
(Please see the attachment)

Reviewer 2 Comments:

Comment #1: Authors should consider adding a section for current widely used methods used for AMD diagnosis in clinical settings. Methods like dilated eye examination, Visual Acuity test, Amsler grid test, fundus photography, OCT, Fluorescein angiography, Electroretinogram, etc., and other methods should be discussed shortly prior to going for the Assessment using the imaging methods section. Even though authors have discussed them in other sections like ERG in section 3, and OCT in section 2, it is advised to start off with a brief overview of these types of diagnostic methods, which are currently widely used in clinical settings.

Reply: We added a Section 2. Clinical Assessment and address methods used for AMD diagnosis in clinical settings per suggestion.

“Early asymptomatic ARMD is typically diagnosed based on the patient's age and a comprehensive dilated eye examination for characteristic signs such as drusen or retinal pigment changes. The progression of AMD to advanced stages invariably involves the foveal region, which develops dense and irreversible scotomas, resulting in retinal function impairment and irreversible vision loss. The Amsler grid can help patients monitor their changes in central vision distortions. Historically, the progression of visual impairment and the estimation of ultimate residual visual function are determined by measuring visual acuity. Standard visual acuity tests, such as best corrected visual acuity (BCVA), do not fully capture the functional impact of atrophic AMD because lesions frequently spare the foveal center in the early stage, causing standard vision charts to falsely indicate that vision is unaffected[16] .Other tests, such as dark adaptation, flicker threshold, and photostress recovery time, are more sensitive than BCVA in detecting early functional loss in AMD[17–19] ;however they are time-consuming and therefore limited in their clinical use. The ophthalmic electrophysiology test, electroretinogram (ERG), can detect the functional abnormalities observed in AMD, such as the early loss of rod photoreceptors and the loss of central and paracentral perimetric sensitivities. Multiple studies have reported the efficacy and accuracy of multi-focal ERG (mfERG) in detecting photoreceptor degeneration and macular function disturbances, which may be beneficial in the early diagnosis and progression of AMD [20–31] . In addition, microperimetry is an automated perimetry system with a non-invasive technique to map retinal sensitivity spatially. Researchers discovered that reductions in retinal sensitivity occur rapidly and precede visual acuity changes in AMD[32–34] . Both mfERG and microperimetry can identify dysfunction in patients with AMD and quantify late-stage progression by measuring local functional deficits in the retina. “

Comment #2: Authors must consider using images of all discussed AMD imaging techniques. This can be done by including images from recently published articles on individual techniques for AMD diagnosis and reproducing them in their review article after getting the necessary permissions from respective publishers.

Reply: We added figures per suggestion.

Comment #3: Line 251: Abbreviate OCT. Acronyms should be abbreviated at their first.

Reply: We revised per suggestion.

Comment #4: Can the reviewers provide a table comparing the pros and cons of all techniques that have been explained in their review article?

Reply: We added a Table to compare the pros and cons of all techniques per suggestion.

Comment #5: Comparatively show the resolution, non-invasiveness, and accuracy/reliability of each of the discussed diagnostic methods. This can be in diagrammatic (preferably) or in table format.

Reply: We included in Table per suggestion.

Comment #6: Can the authors give a short note on the effective utilization of these techniques being relied on in clinical scenarios? Adding this in the end section will be of more useful for readers

Reply: We thank you for the comment; we added in Section 4, paragraph 2, line 458-466:” For AMD eyes with no clinical indication of active or regressed CNV, a baseline CFP is recommended, while an ICG-A test may be used in some instances. Throughout the follow-up, FA may be required at any visit to assess for NV. For patients with nAMD, FA is recommended at baseline and selected follow-up visits while an ICG-A can also be used for visualization of the vasculature. For optimal detection and measurement of late AMD atrophy and associated characteristics of MA, multimodal imaging is advised in clinical settings. Imaging include CFP, FAF, confocal NIR, and high-resolution OCT volume scans images or OCTA should be acquired at regular intervals throughout the study to detect, quantify, and monitor progression of atrophy.”

Comment #7: It is true that among the different versions of OCT, OCTA is a go-to when assessing for eye imaging. There have been several research with different types of OCT like PS-OCT, Phase OCT, ODT, and other variations useful in the diagnosis of AMD. Authors should address these variations and their usefulness in different scenarios.

Reply: We address different types of OCT like PS-OCT, Phase OCT, ODT in the diagnosis of AMD in Section 3.4. , paragraph 5 , line 384-399 “Recent polarization-sensitive optical coherence tomography (PS-OCT) was utilized to estimate the melanin content of RPE, with the surface of the atrophy area having high entropy (low melanin).[128] In contrast, standard imaging tools, such as CPF or FAF, are incapable to determine the degree of RPE pigmentation. Entropy values in PS-OCT are suggested as a detective tool to assess the degree of RPE pigmentation and, hence, the health of RPE; however, further research is needed to validate this. A combined OCT imaging system and laser Doppler flowmetry, optical Doppler tomography (ODT), allows the quantitative imaging of fluid flow in a highly scattering medium.[129] In addition to traditional structural OCT, ODT evaluates tissue function in AMD eyes with possible abnormal ocular circulation. However, clinical applications of this functional OCT are still in their early stage of development. Similarly, an angiogram without fluorescent dye injection, phase contrast optical coherence tomography (PC-OCT), was developed to image retinal microvasculature.[130] In the atrophic area of GA, PC-OCT angiographic image showed large choroidal vessels and loss of the overlying superficial choriocapillaris. However, there is still no consensus about if patients with GA truly have an absence of flow or only a reduction in flow[131] . Hence, these findings warrant further clarification.”

Comment #8: Lines: 319 to 320, this is misleading and wrong, it is not like SS-OCT offers a larger scan area. This depends on the optics used in the system. Besides, SS-OCT may be introduced later on, but it is not a new technique. It is a variation that has been well-reported.

Reply: we revised in Section 3.4. , paragraph 4 , line 376-383: “In contrast to SD-OCT, which employs an interference spectrum acquired through spectral splitting and a low-coherence light source, SS-OCT typically utilizes a broad band sweep source in which the wavelength of the light source fluctuates over time.[124] The longer wavelengths of SS-OCT enable better penetration into the choroid, and excellent reproducibility and repeat ability of choroidal thickness measurements [125,126] . Using SS-OCT imaging to detect hypertransmission into the choroid, the cRORA[127] and iRORA[118] can be identified via an en face slab with boundaries beneath the RPE.”

Comment #9: Enhanced Depth in OCT can be achieved mu multiple methods using variations and different optical setups. Reporting it as one specific variation of OCT can be misleading to readers.

Reply: we deleted the misleading description per suggestion.

Comment #10: In section 4, can the authors include more in details of the different diagnostic methods, future trends, and expected advancements that can greatly help doctors to better identify and early diagnosis of AMD?

Reply: We included more in details of using machine learning algorithms such as deep learning model to better identify and early diagnosis of AMD in section 4, paragraph 1, line 444-457 “In contrast, machine learning algorithms can serve as a very useful tool in the automated quantification of pathologic characteristics[156]. A study indicated that a deep learning model to classify atrophy on FAF imaging can accurately differentiate atrophy caused by GA from its masqueraders - inherited macular dystrophies[157] . Recently, machine learning, specifically deep learning techniques have been applied to FAF and OCT imaging to detect and classify GA [157–160] . More recently, a deep learning model using convolutional neural networks was developed and validated for segmentation of the 13 most common features related to early and late nAMD[161]. In addition, the application of deep learning algorithms to the prediction of progression to nAMD has favorable results[162] .The utilization of artificial intelligence such as deep learning model in automated OCT analysis could potentially facilitate the early detection and predict progression of AMD[163] , thereby expanding the future therapeutic window of opportunity. As the deep learning model further develops, it may be able to early detect and predict the risk of developing MA and provide customized treatment strategies.”

Comment #11: Also, under section 4, can the authors include the use of Deep-learning and other Machine learning methods that are being currently explored and reported by researchers and doctors alike for a better understanding of different imaging diagnostics and for early diagnostics? Also, even a short mention of how AI-assisted image analysis and machine learning can be expected to further advance understanding and early diagnosis of AMD. 

Reply: We thank you for the comment; we replied in comment #10.  

Round 2

Reviewer 1 Report

Comments and Suggestions for Authors

The authors have done a good job re-organizing their manuscript, going through the different imaging modalities in details, and adding figures with multimodal imaging of GA.

However, the authors still have not discussed imaging neovascular AMD which is nearly a third of the cases with "late AMD" and there are still extensive description of drusen types and patterns on imaging, which are not forms of "late AMD" as the title would suggest. I understand - as the authors state in their response - that drusen are a risk for developing late AMD but you do not define a certain stage of a disease to review (imaging Late AMD) and then refrain from discussing a big part of that topic (neovascular AMD & how to detect it) due to word count limitations, and yet go into exhaustive details of how to image preceding stages (early-intermediate) AMD. If the authors wish to only discuss drusen and atrophy, then the title and whole manuscript should be clear about the purpose of the review, "imaging macular atrophy" for example or "imaging dry-type AMD." All three figures are of macular atrophy. Also, if word count is a concern, there are multiple sections that go into details outside the scope of the review, e.g. prevalence details of AMD and its distribution among ethnicities, regions.

The title defines the scope of the review as "recent advances" but the authors discuss modalities that have mostly been there for decades. Newer modalities for imaging MA, such as adaptive optics, microperimetry..etc., are not even mentioned. Home monitoring OCT devices are on the forefront of research and some are nearing market distribution, but the authors do not discuss those as well.

The authors still do not provide a clear algorithm for a reader who is more than likely a clinician who wants to know in summary, which imaging modality should I use? and when? and how often? and when to suspect neovascular disease?

There are still some minor arrangement issues in the manuscript, for example the figure labels (letters) are not clear in all figures.

Author Response

We thank you and reviewer for the valuable comments. We have revised the manuscript accordingly and provided the following point-to-point responses:

Reviewer 1 Comments:

Comment #1: The authors have done a good job re-organizing their manuscript, going through the different imaging modalities in details, and adding figures with multimodal imaging of GA. However, the authors still have not discussed imaging neovascular AMD which is nearly a third of the cases with "late AMD" and there are still extensive description of drusen types and patterns on imaging, which are not forms of "late AMD" as the title would suggest. I understand - as the authors state in their response - that drusen are a risk for developing late AMD but you do not define a certain stage of a disease to review (imaging Late AMD) and then refrain from discussing a big part of that topic (neovascular AMD & how to detect it) due to word count limitations, and yet go into exhaustive details of how to image preceding stages (early-intermediate) AMD. If the authors wish to only discuss drusen and atrophy, then the title and whole manuscript should be clear about the purpose of the review, "imaging macular atrophy" for example or "imaging dry-type AMD." All three figures are of macular atrophy. Also, if word count is a concern, there are multiple sections that go into details outside the scope of the review, e.g. prevalence details of AMD and its distribution among ethnicities, regions.

Reply: We deleted prevalence details of early AMD and distribution among ethnicities, regions.  In addition, we revised the title to “Recent Advances in Imaging Macular Atrophy for Late-Stage Age-Related Macular Degeneration” to avoid confusion per suggestion.

Comment #2: The title defines the scope of the review as "recent advances" but the authors discuss modalities that have mostly been there for decades. Newer modalities for imaging MA, such as adaptive optics, microperimetry..etc., are not even mentioned. Home monitoring OCT devices are on the forefront of research and some are nearing market distribution, but the authors do not discuss those as well.

Reply: We added “New modalities for imaging MA” in Section 4.2. line  460-505 to discuss “Microperimetry, Adaptive Optics, Home-based OCT” per suggestion.

Comment #3: The authors still do not provide a clear algorithm for a reader who is more than likely a clinician who wants to know in summary, which imaging modality should I use? and when? and how often? and when to suspect neovascular disease?

Reply: We edited Section 4.1. Imaging algorithm in line 429-441 to provide algorithm including when, which, and how often to use imaging modalities to detect AMD eyes with and without neovascular or monitor MA progression.

Comment #4: There are still some minor arrangement issues in the manuscript, for example the figure labels (letters) are not clear in all figures.

Reply: We edited all the figure labels and letters accordingly.

Reviewer 2 Report

Comments and Suggestions for Authors

1) Did the authors change the title? "Late-stage" to "Late", was this a mistake, or was it intentional? If intentional, it doesn't make sense.

2) Scale Bars are missing in Fig.1, also explanation i.e., figure descriptions which give the details of each part of the figures (A- E) are missing. This should be done for all figures and tables. The general format is "Figure Number. Figure Title. Figure Caption". None of the figures have figure/image labels for the bottom right photographs or images (D or E).

3) Authors should be more careful when making the review response file and edited text in the manuscript. The line numbers given in review responses are not correct, and highlights are missing in the manuscript text (Mentioned as Response to Reviewer's Comment 8).

Comments on the Quality of English Language

Minor English proficiencies need to be addressed. Articles are missing.

Author Response

We thank you and reviewer for the valuable comments. We have revised the manuscript accordingly and provided the following point-to-point responses:

Reviewer 2 Comments:

Comment #1: Did the authors change the title? "Late-stage" to "Late", was this a mistake, or was it intentional? If intentional, it doesn't make sense.

Reply: Thank you for the typo correction. We corrected the title and edited it per reviewer 1's suggestion to “Recent Advances in Imaging Macular Atrophy for Late-Stage Age-Related Macular Degeneration.”

Comment #2: Scale Bars are missing in Fig.1, also explanation i.e., figure descriptions which give the details of each part of the figures (A- E) are missing. This should be done for all figures and tables. The general format is "Figure Number. Figure Title. Figure Caption". None of the figures have figure/image labels for the bottom right photographs or images (D or E).

Reply: We added scale bars in Figure 1B-E. Figure format was uploaded unsuccessfully in revision 1. We will reupload in this revision. We added image labels for the bottom right photographs or images (E).

Comment #3:Authors should be more careful when making the review response file and edited text in the manuscript. The line numbers given in review responses are not correct, and highlights are missing in the manuscript text (Mentioned as Response to Reviewer's Comment 8).

Reply: We update and make sure line numbers given in review responses are correct. In addition, we will upload the correct version of manuscript text in this revision.

Round 3

Reviewer 1 Report

Comments and Suggestions for Authors

The review now reads better and the title makes more sense. Some final edits are, however, required:

- Section 4 title, suggest a change from "ongoing trends in management & research" to "Current Approach and Future Directions"

- Lines 404-406 add no value and can be deleted.

- Line 411: please add OCT-A to angiography methods of detection nAMD.

- Lines 412-415: For nAMD, please change interval of follow up from "6-12 months" to "1-3 months" as this makes more sense and is the most recent recommendation. Please also add a citation to this sentence, I suggest you use the latest recommendation (Fouad et al. Pathways of fluid leakage in age-related macular degeneration. Retina. 2023).

- For figure 2, the authors label the eye as atrophy associated with nAMD. To me, this seems like degenerative microcysts associated with the underlying RPE atrophy (discussed in the Retina editorial I mentioned above), and not intraretinal fluid due to neovascular activity. Do the authors have dye-based imaging that shows a neovascular membrane?

- Figure labels (A, B, C...) are still distorted and need to be fixed.

- Some minor language errors need to be modified, e.g.,

line 442, punctuation after "sensitivity" and before "reductions"

line 443, space between "has" and "eye"

Comments on the Quality of English Language

Mentioned above.

Author Response

We thank you reviewer for the valuable comments. We have revised the manuscript accordingly and provided the following point-to-point responses:

Comment #1:  Section 4 title, suggest a change from "ongoing trends in management & research" to "Current Approach and Future Directions"

Reply: We revised Section 4 , line 423 title to "Current Approach and Future Directions". 

Comment #2: Lines 404-406 add no value and can be deleted.

Reply: We agreed and deleted that sentence per suggestion.

Comment #3: Line 411: please add OCT-A to angiography methods of detection nAMD.

Reply: We edited in line 430 to “Angiography (FA, ICG-A and OCTA) maybe indicated in the event that a neovascular process is suspected in follow-up.”

Comment #4: Lines 412-415: For nAMD, please change interval of follow up from "6-12 months" to "1-3 months" as this makes more sense and is the most recent recommendation. Please also add a citation to this sentence, I suggest you use the latest recommendation (Fouad et al. Pathways of fluid leakage in age-related macular degeneration. Retina. 2023).]\

Reply: We agreed and changed the interval of follow up from "6-12 months" to "1-3 months" in line 434. We cited this sentence with the suggested article.

Comment #5: For figure 2, the authors label the eye as atrophy associated with nAMD. To me, this seems like degenerative microcysts associated with the underlying RPE atrophy (discussed in the Retina editorial I mentioned above), and not intraretinal fluid due to neovascular activity. Do the authors have dye-based imaging that shows a neovascular membrane?

Reply: We do not have an image of this patient's neovascular membrane. To avoid confusion, we have included a dye-based imaging image illustrating a neovascular membrane from a different patient in this reply and replaced the images in Fig. 2 from this particular patient.

Comment #6: Figure labels (A, B, C...) are still distorted and need to be fixed.

Reply: We fixed figure labels accordingly.

Comment #7: Some minor language errors need to be modified, e.g., line 442, punctuation after "sensitivity" and before "reductions" line 443, space between "has" and "eye"

Reply: We thank you for the correction of language errors, which we modified accordingly.